# Overexpression of Nucleolin and Associated Genes in Prostate Cancer

**DOI:** 10.3390/ijms23094491

**Published:** 2022-04-19

**Authors:** Virginie Firlej, Pascale Soyeux, Maya Nourieh, Eric Huet, Fannie Semprez, Yves Allory, Arturo Londono-Vallejo, Alexandre de la Taille, Francis Vacherot, Damien Destouches

**Affiliations:** 1Univ Paris Est Creteil, TRePCa, F-94010 Creteil, France; virginie.firlej@u-pec.fr (V.F.); pascale.porte@u-pec.fr (P.S.); huet@u-pec.fr (E.H.); alexandre.de-la-taille@aphp.fr (A.d.l.T.); vacherot@u-pec.fr (F.V.); 2Department of Pathology, Institut Curie, F-92210 Saint-Cloud, France; maya.nourieh@curie.fr (M.N.); yves.allory@curie.fr (Y.A.); 3SPPIN—Saints-Pères Paris Institute for the Neurosciences, Université de Paris, CNRS, F-75006 Paris, France; fannie.semprez@u-paris.fr; 4Institut Curie, PSL Research University, CNRS UMR 144, F-75005 Paris, France; 5Institut Curie, PSL Research University, CNRS UMR 3244 « Telomeres and Cancer », F-75005 Paris, France; jose-arturo.londono-vallejo@curie.fr; 6AP-HP, Hôpital Henri-Mondor, Service Urologie, F-94010 Creteil, France

**Keywords:** prostate cancer, nucleolin, biomarker

## Abstract

Prostate cancer (PCa) is the second most frequent cancer and the fifth leading cause of cancer death in men worldwide. If local PCa presents a favorable prognosis, available treatments for advanced PCa display limiting benefits due to therapeutic resistances. Nucleolin (NCL) is a ubiquitous protein involved in numerous cell processes, such as ribosome biogenesis, cell cycles, or angiogenesis. NCL is overexpressed in several tumor types in which it has been proposed as a diagnostic and prognostic biomarker. In PCa, NCL has mainly been studied as a target for new therapeutic agents. Nevertheless, little data are available concerning its expression in patient tissues. Here, we investigated the expression of NCL using a new cohort from Mondor Hospital and data from published cohorts. Results were then compared with NCL expression using *in vitro* models. NCL was overexpressed in PCa tissues compared to the normal tissues, but no prognostic values were demonstrated. Nine genes were highly co-expressed with NCL in patient tissues and tumor prostate cell lines. Our data demonstrate that NCL is an interesting diagnostic biomarker and propose a signature of genes co-expressed with NCL.

## 1. Introduction

Prostate cancer (PCa) is currently the second most common malignancy and the fifth leading cause of cancer death worldwide with 1.3 million new cases and 360,000 deaths per year estimated in 2018 [1]. While PCa is frequently curable when it is diagnosed early, advanced PCa represents a significant source of mortality and morbidity. As prostate tumor cells are dependent on androgens for their survival and growth, the gold standard therapy is then androgen deprivation therapy (ADT). Unfortunately, after an initial efficient clinical response, tumors relapse within 1–3 years, leading to castrate-resistant prostate cancer (CRPC). Since the last decade, new therapies are available for CRPCs, including new androgen receptor pathway inhibitors (ARPIs, such as the abiraterone acetate, enzalutamide, apalutamide, or darolutamide), chemotherapies (docetaxel and cabazitaxel), and radionuclides (radium-223 or Lutenium-177) [2,3,4,5,6,7,8,9]. Nevertheless, due to primary or acquired resistances, all these treatments remain palliative, leading to patient death within 9–30 month after progression to CRPC [10,11]. Thus, research efforts are still needed to discover new targets and to provide new therapeutic solutions for advanced PCa.

Nucleolin (NCL) is a ubiquitous protein and one of the most abundant phosphoproteins of the nucleolus. NCL is able to shuttle from the nucleolus to the nucleus, cytoplasm and cell surface [12,13]. Depending on its localization, NCL can display many functions in ribosome biogenesis, DNA repair, epigenetic modifications, genome stability, cell division and survival, angiogenesis and lymphangiogenesis, epithelial–mesenchymal transition, and stemness [14]. NCL is overexpressed in cancer cells from many histological origins [13]. NCL is overexpressed at the cell surface in tumor cells and in activated endothelial cells [15,16,17]. It can interact with several proteins, including FAS ligand, ErbB1 and RAS, pleiotrophin and Midkine, or VEGF, thus regulating cancer cell survival and proliferation, and angiogenesis [18,19,20,21,22,23,24].

The first study of NCL in PCa has demonstrated an association with a decrease of its expression and phosphorylation in tumor cell proliferation under androgen deprivation [25]. In CRPC cell lines, NCL was mainly localized at the cell surface compared to hormone naïve cell lines [26]. Cell surface nucleolin has been described as a novel receptor for HGF, leading to an increase in adhesion and migration of PC3 and C4-2 prostate cell lines [27]. A role for NCL in epithelial–mesenchymal transition has been recently demonstrated. NCL acetylation and methylation by the CITED2/PRMT5 and p300 complex has led to the translocation of nucleolin from the nucleus to the cytoplasm, promoting both epithelial–mesenchymal transition and cell migration in PC3 and DU145 prostate cell lines [28]. The CITED2-NCL pathway has been proposed as a potential target for prostate cancer metastasis therapy.

In PCa, the majority of studies on NCL investigate the anti-tumor activities of targeting molecules or decipher their mechanisms of action using *in vitro* models. Recently, NCL inhibition by a single-chain fragment-variable (scFv, 4LB5) resulted in the reduced proliferation and migration of tumor prostate DU145 and PC3 cells [29]. As for the 4LB5 scFv, the aptamer AS1411 and its formulated nanoparticles, the pleiotrophin P11-136 peptide and the multivalent pseudopeptides NucAnts, have demonstrated a growth inhibition of PCa cells [30,31,32,33,34,35,36,37].

Although NCL functions and its targeting have been mainly studied in prostate cancer cell lines, its expression in human PCa has not been clearly established. In the present study, we investigated *NCL* mRNA and protein expressions in a large cohort of prostate adenocarcinoma established at Mondor Hospital and in public databases. We show that *NCL* is highly expressed in PCa compared to normal tissue and its expression is overexpressed in castration-resistant prostate cancer (CRPC), although no prognostic value has been demonstrated. Several genes associated with cell division and ribosome biogenesis are co-expressed with *NCL* in tissues and cellular models, highlighting the role of this protein in prostate cancer.

## 2. Results

### 2.1. Baseline Patient Characteristics

The PAIR prostate (PP) cohort includes 9 normal peritumoral tissues, 116 hormono-naive prostate cancer (HNPC) tissues, and 13 castrate-resistant prostate cancer (CRPC) tissues. HNPC clinical, biochemical, and pathologic characteristics are described in Table 1.

### 2.2. Nucleolin Is Overexpressed in Prostate Cancer

The mRNA expression of *NCL* in PCa tissues was first analyzed using mRNA from the PP cohort. The mRNA was extracted from FFPE of PCa tissues and its expression was analyzed using a GeneChip^®^ Human Transcriptome array 2.0 (Affymetrix, Thermofisher scientific, Courtaboeuf, France). The expression of *NCL* was significantly higher in prostate tumors compared to normal tissues (*p* = 0.0031, Figure 1a). This overexpression was confirmed with paired samples from normal peritumoral and tumor tissues for the same patients with a higher significant difference (*p* = 0.0005, Figure 1b).

In order to validate the results obtained in our PP cohort, the expression of *NCL* in PCa was evaluated by the analysis of the available TCGA and GTEx cohorts for PCa patients using the XENA platform (including 100 normal tissues, 52 solid normal tissues (peritumoral normal tissue), and 495 HNPC) (Figure 1c). *NCL* was significantly overexpressed in tumors compared to solid tissues (*p* < 0.001) and with a higher difference when compared to normal tissue (*p* < 0.001). Our PP cohort exhibits samples from PCa at different stages (HNPC and CRPC). *NCL* expression was significantly higher in CRPC compared to HNPC (*p* < 0.001), suggesting a role of NCL in the castration-resistance of prostate cancer (Figure 1d).

In order to validate the results at the protein level, the expression of NCL was then analyzed by IHC using FFPE samples. A slight, and mainly nuclear, staining was observed in peritumoral tissues (Figure 1e). NCL expression was highly increased in tumors with nuclear but also cytoplasmic localizations. The NCL staining analysis suggested an increased expression of the protein in CRPC compared to HNPC. Nevertheless, the quantification of NCL using quick score did not demonstrate a difference between HNPC and CRPC at the protein level, probably due to the high heterogeneity of staining in tumors as well as the low CRPC samples number (Figure 1f).

These results demonstrate the overexpression of NCL in tumor tissues compared to the normal ones and suggest an increased expression of NCL in CRPC compared to HNPC.

### 2.3. NCL Expression Is Not a Prognosis Biomarker of PCa

To determine if the value of mRNA *NCL* expression could improve the detection of high-risk prostate cancer, we evaluated its expression in the PP cohort using the associated clinic parameters. The expression was first analyzed with the Gleason score (GS). No significant difference in *NCL* expression was observed between low (6 and 3 + 4) and high (4 + 3, 8 and 9) GS (Figure 2a).

Prognostic risks can be assessed as low (GS 6/7 (3 + 4) and TNM stage pT2), intermediate (GS (4 + 3) and TNM stage pT2 or pT3a), or high (GS 8/9 and/or TNM stage pT3b/4a). No significant difference was observed between the different risk groups (Figure 2b), confirming the first result taking into account only the GS. Using HNPC samples, the survival of patients with low versus high expression was analyzed. The cut-off expression was determined by averaging the relative expressions of *NCL* in HPNC patients. Again, no significant difference was observed (Figure 2c). A similar analysis was performed by querying the TCGA gene expression database. Here, a slight significant difference has been observed between low and high GS (Figure 2d) and between low- and high-risk groups (Figure 2e).

Altogether, although there was an increased level of *NCL* mRNA in PCa tissues, these results do not present evidence that its expression could be a powerful prognostic biomarker.

### 2.4. Genes Positively Correlated with NCL Have Function in Cell Division and Ribosome Biogenesis

We found that *NCL* was overexpressed in prostate tumor tissues. To clarify its potential function in cancer development/progression, the mRNA expression data of 16,200 genes were analyzed in the PP cohort. The expression of genes that could be overexpressed with *NCL* was analyzed using Spearman correlations. Genes positively co-expressed with *NCL* displaying a high Spearman correlative score (r > 0.55) were submitted for DAVID software analysis. Among the main enriched biological themes (GO terms), the genes positively co-expressed with *NCL* were associated with cell cycle and ribosome biogenesis (Figure 3a).

In parallel, the potential connection of these genes with *NCL* was assessed using the GIANT analysis. Among all these genes, the seven genes that presented the higher GIANT score also displayed a high Spearman correlation score (superior to 0.5) in both our PP cohort and the available TCGA cohort: *HSPD1*, *PAICS*, *CCT5*, *NPM1*, *SERBP1*, *GART*, and *PA2G4* (Table 2).

The correlations of the expression of *NCL* and these seven genes are represented in Figure 4a–g. The very low correlation of *NCL* and *CSP1* expressions is given as a negative example (Figure 4h). These correlations of expression are highlighted with a heatmap representation in Figure 4i.

Using different cohorts and analysis tools, these results suggest a strong link between *NCL* and the positively co-expressed genes *HSPD1*, *PAICS*, *CCT5*, *NPM1*, *SERBP1*, *GART*, and *PA2G4*. This potential functional link should be evaluated in further studies.

### 2.5. Tumor Cell Lines Highly Expressed NCL

Several cell lines are usually used as in vitro models for PCa studies. We next evaluated NCL expression in a panel of these cell lines, including tumoral and non-tumoral cell lines. LNCaP and VCaP were used as androgen-dependent cell lines; LNCaP-AI, DU145, PC3, and 22Rv1 as androgen-independent cell lines; LNCaP-NE as a neuro-endocrine-like cell line; and WPMY1 (immortalized prostate myofibroblast) and PNT2 (immortalized prostate epithelial cells) as non-tumoral cell lines.

The mRNA expressions were analyzed with the mRNA from PCa tissues using GeneChip^®^ Human Transcriptome array 2.0 (Affymetrix, Thermofisher scientific, Courtaboeuf, France). *NCL* mRNA expression was quite high in all tested cell lines and can be compared to the level of expression measured in tumor tissues (Figure 5a). Only two cell lines showed a lower signal: the LNCaP-NE cells and the DU145 cells. The NCL protein expression was evaluated by Western blot in triplicate to quantify the different levels of expression using Image J software. The protein expression was well correlated with mRNA expression with a high expression in most of the cell lines studied, except the LNCaP-NE and DU-145 cells, in accordance with their *NCL* mRNA level (Figure 5b,c).

Using the transcriptomic data of the cell lines, we evaluated the expression profiles of NCL with the seven genes previously described to be co-overexpressed in PCa tissues. As for patient samples, Spearman correlation scores were assessed and were represented in a heatmap (Figure 5d). The HSPD1, PAICS, CCT5, NPM1, SERBP1, GART, and PA2G4 mRNA expressions were well correlated to the NCL expression as observed with PCa tissues. As for NCL expression, the seven genes displayed a lower expression in the LNCaP-NE and DU-145 cell lines.

These data demonstrate that the different cell lines could be used in the future as relevant in vitro models to study the functional links between NCL and these genes.

## 3. Discussion

Current therapies for advanced PCa are mainly focused on the inhibition of the AR pathway. Nevertheless, despite efficient initial responses, resistance will occur, leading to patient death. One of the major challenges in PCa remains the discovery of new targets in order to design more effective therapies.

Nucleolin is a multifunctional protein localized in different compartments of the cell (nucleolus, nucleus, cytoplasm, and cell surface). Its global expression is strongly increased in cancer cells and NCL has been proposed as a tumor marker for cancers from many histological origins, such as cervical carcinoma, pediatric intracranial ependymoma, pancreatic ductal adenocarcinoma, glioma, gastric, hepatocellular, lung, colorectal, and endometrial carcinoma [15,17,34,38,39,40,41,42,43,44]. In this study, using a new cohort of patients from Mondor Hospital and publicly available data (TCGA and GTEX), we demonstrated that *NCL* is also overexpressed in PCa tissues compared to non-tumoral tissues and can be considered as a tumor biomarker in PCa.

Correlations between *NCL* expression and prognosis issue or clinicopathological features are often investigated. High levels of *NCL* expression are associated with poor prognosis and decreased overall survival in hepatocellular carcinoma, pediatric intracranial ependymoma, and pancreatic ductal carcinoma [17,39,43]. NCL localization is also considered as a prognosis marker. Indeed, it has been demonstrated in small-cell lung cancer, pancreatic ductal carcinoma, and gastric cancer that nuclear localization is associated with a good prognosis factor, whereas prognosis is considered worse with high-level expression of NCL in the cytoplasm [40,41,43]. The association of *NCL* expression with clinicopathological features is more controversial. *NCL* expression was associated with the grade and clinical stage in glioblastoma, hepatocellular carcinoma, and non-small-cell lung cancer [15,43], whereas no significant difference was observed in pancreatic ductal carcinoma and gastric cancer [40,41]. Two recent studies investigated *NCL* expression with clinicopathological features. *NCL* expression was decreased in prostate tumor cells infiltrating seminal vesicles compared to prostate-confined tumor cells and no difference was observed between Gleason scores (GS) of three and four [45]. Sheetz et al. investigated *NCL* expression in tumor tissues using the available database TCGA. In this study, increased *NCL* mRNA expression was associated with Gleason scores (8–10 > to 6–7), and with biochemical recurrence and metastases [29]. This result was confirmed using the TCGA cohort.

Based on our study, we showed for the first time an increased expression of *NCL* mRNA in CRPC patients compared to HNPC. Nevertheless, no significant difference of NCL expression between low (6 and 3 + 4) and high (4 + 3, 8, and 9) GS has been observed. The difference between studies, is that we differentiate the 3 + 4 and 4 + 3 GS. Indeed, two different ISUP scores (grade group two vs. three) can discriminate the aggressiveness of prostate cancer: GS 3 + 4 is considered low risk (group 2) whereas GS 4 + 3 correspond to a higher risk with worse diagnostics and survival [46]. A slight but significant increase in *NCL* level was observed in the high- versus the low-risk group in the TCGA database but no difference was observed using data of the PP cohort. However, this difference is related to two outlier patients with a very high expression of *NCL* with a GS of seven (3 + 4). Moreover, using the PP cohort, no significant difference was shown in progression-free survival. NCL seems to remain controversial for its use as a prognostic value, but, as evidenced in the increase in tumor cells versus adjacent non-neoplasia tissues, it could be a promising diagnostic biomarker.

*NCL* was successfully detected in prostate tumor cells in post-digital rectal-massage urine [47]. Currently, extracellular vesicles (EVs) are highlighted in the biomarker area, since their content reflects the one of the cell of their origin and they are present in biological fluids such as plasma/serum or urine [11]. Our preliminary results showed that this protein was also detected in EVs secreted by prostate tumor cell lines (data not shown). It is then tempting to speculate that NCL could be used as an EV biomarker in biological fluid.

We then investigated the NCL-positive co-expressed genes in our PP cohort. Among the genes with the highest Spearman correlation scores in the PP and TCGA cohorts, seven were also predicted to have a connection with NCL using GIANT software. Of the HSPD1 encoded for the chaperon heat shock proteins, 60 are involved in PCa development [48]. The de novo purine biosynthetic gene PAICS has been proposed as a therapeutic target in PCa [49]. SERBP1 and encode serpine1-binding protein 1 have been shown to be overexpressed in PCa and were significantly associated with tissue metastasis and high GS [50]. Proliferation-associated protein 2G4 (or EBP1), encoded by the *PA2G4* gene, regulates AR translation in CRPC [51]. Finally, nucleophosmin-1 (NPM1), as NCL, is a phosphoprotein overexpressed in PCa and especially CRPC specimens and regulates RA activity [52]. No studies in prostate cancer have investigated the roles or the expressions of the T-complex protein 1 subunit epsilon, encoded by the *CCT5* gene, or the trifunctional purine biosynthetic protein adenosine-3, encoded by the *GART* gene. If these genes have been demonstrated to be overexpressed in PCa, no studies have investigated the links between these genes and NCL. The investigation of NCL expression in normal and tumoral prostate cell lines showed a high expression level. The positive correlation of the seven genes with NCL was confirmed in different cell lines mimicking PCa evolution. These results demonstrate that these cell lines can be used as good in vitro models for studying the mechanistic links between NCL and these seven genes.

Targeting NCL has been demonstrated as an efficient strategy to inhibit tumor growth using several compounds, such as the aptamer AS1411, the multivalent pseudopeptides NucAnts, the F3 peptide, and endostatin [16,36,37,53,54,55]. AS1411, also named AGRO100, is an aptamer that binds NCL and induces tumor cell death [56]. Its efficiency, optimization, and mechanisms of action were mainly studied in vitro in tumor prostate cell lines. Using DU145 cells, AS1411 was able to bind and form a complex with the nuclear factor-kappaB essential modulator (NEMO) and nucleolin leading to the inhibition of the nuclear factor-kappaB (NF-KappaB) activity [30]. To understand the differential effects of AS1411 between resistant and sensitive cells, its mechanism of internalization has been studied in DU145 and Hs27 cell lines. AS1411 was able to bind and to be internalized in sensitive cells (DU145) and resistant cells (Hs27) but with different mechanisms of internalization [31]. In sensitive DU145 cells, AS1411 was internalized by micropinocytosis, whereas it was internalized by classical endocytosis in Hs27-resistant cells. NCL was necessary to induce the micropinocytosis of AS1411. A more recent study has demonstrated that NCL is involved in micropinocytosis through the EGFR/Rac1 pathway in LNCaP and DU145 cell lines [57]. *Nucleolin* has also been proposed as a marker to identify the circulating tumor cells of prostate cancer [58] and as a target for prostate cancer characterization by optical contrast-enhanced photoacoustic imaging using F3 peptide in nanoparticles [34]. The antitumoral activity of N6L in PCa was previously demonstrated with a promising efficiency, especially for CRPC [52]. In a previous study, we showed a correlation between NCL expression and N6L antitumoral activity in pancreatic ductal carcinoma [17]. This correlation was not observed in PCa cell lines, probably in relation to a high basal level in all cell lines (data not shown).

In conclusion, NCL seems to be a key player in cancer. Here, we demonstrated that NCL is a diagnostic marker for PCa but its use as a prognostic biomarker remains questionable, such as for other cancers. All these results confirm NCL as a promising target for anti-cancer therapy.

## 4. Materials and Methods

### 4.1. Human Prostate Cancer Specimens

Prostate tissue samples were collected as part of an Institutional Review Board-approved protocol at Henri Mondor Hospital in France. In this cohort, 129 PCa tissue samples were collected, including 116 samples from the radical prostatectomy of patients that did not receive prior hormone treatment at the hospital (hormone-naïve prostate cancer; HNPC) and 13 tissues were collected by transurethral resection from castrate-resistant prostate cancer (CRPC) patients. Among these tissues, 9 HNPC specimens derived from normal peritumoral tissues were paired with tumoral tissues. Demographic, clinical, and pathological criteria were prospectively collected in a database and retrospectively reviewed. Tumors were classified based on histomorphology by our genitourinary pathologist Y. Allory based on TNM 2018. Tumor specimens are classified into low-risk, intermediate-risk, and high-risk groups using an adaptation of D’Amico’s classification, which does not take in account the PSA rate but only the histologic data on the basis of the Gleason score and TNM features, as performed previously [59]. Tumors with Gleason score 6/7 (3 + 4) and TNM stage pT2 were classified as low risk. Tumors with Gleason score 8/9 and/or TNM stage pT3b/4a were defined as high risk. Tumors classified as pT3a, or pT2 with Gleason score 7 (4 + 3) were considered as intermediate.

The Cancer Genome Atlas (TCGA) cancer 2015 and 2018 datasets based on TCGA Research Network (http://cancergenome.nih.gov, 19 January 2022) were queried for mRNA expressions in prostate cancer using cBioportal and GTEx data using Xena platform [60].

### 4.2. RNA Microarray and Transcriptomic Data

Total RNA was purified from the frozen tissues using the miRNeasy kit (Qiagen, les Ulis, France) and transcriptome profiles were generated from HNPC (n = 54) and CRPC (n = 13) prostate cancer tissues and performed using GeneChip^®^ Human Transcriptome array 2.0 (ThermoFisher Scientific, Coutaboeuf, France). Data were analyzed by Genosplice company as previously described [61,62] (Genosplice, Paris, France).

### 4.3. Bioinformatic Analysis

Correlation of mRNA expression between NCL and the other genes was performed using a Spearman test.

To identify enriched biological themes (GO terms) corresponding to genes co-regulated with NCL, the integrated regulatory network from Perl script DAVID was used. This analytic tool serves for researchers to understand the biological meanings from a large list of genes or proteins [63]. The positively correlated expressed genes with a correlation Spearman score superior to 0.55 were submitted for DAVID analysis. For functional annotation of genes in the interaction network, we identified the over-represented gene ontology (GO) categories in biological processes by DAVID with the Benjamini *p* < 0.01 and a fold enrichment > 1.7. Thus, 67 genes on 16,202 were selected.

Prediction of gene network linked to NCL was performed using the GIANT (Genome-scale Integrated Analysis of gene Networks in Tissues) software (http://giant-v2.princeton.edu, 19 January 2022). Search was performed in prostate gland with a gene max = 29 and a confidence score = 0.5.

### 4.4. Human Prostate Cancer Tissues and Immunohistochemistry

Immunohistochemistry analysis was performed as previously described with minor modifications [52]. Formalin-fixed paraffin-embedded (FFPE) tissues were sectioned at 5 µm thickness, deparaffinized, and rehydrated. Antigen were unmasked by heat retrieval with pH 9 EDTA buffer for 15 min and endogenous peroxidase activity was inactivated with a 3% hydrogen peroxide solution for 10 min. Tissues were then immuno-stained overnight at 4 °C with anti-nucleolin antibody (sc-8031, Santa-Cruz Biotechnology, Heidelberg, Germany, 1:50) and diluted in antibody diluent (Zytomed, Berlin, Germany). Immuno-complexes were revealed using the anti-mouse Polink HRP mouse kit and the DAB substrate (Diagomics, Blagnac, France) with an incubation of 30 min at room temperature. Tissues were then stained with hematoxylin and dehydrated. Slides were mounted using Eukitt medium. Protein expression was scored as null (0), weak (1), moderate (2), and strong (3), and the percentage of tumor cells stained was noted. The multiplication of the score and the percentage of stained tumor cells gave the quick score (between 0 and 300). Analysis was performed separately by one genitourinary pathologist (MN) and by two scientific investigators (DD and VF).

### 4.5. Cell Culture

All cell lines were purchased from ATCC (American Type Culture Collection/LGC Promochem, Molsheim, France). The 22Rv1, DU145, PC3, and PNT2 cells were cultured in RPMI supplemented with 10% of fetal bovine serum (FBS). LNCaP were cultivated in RPMI, 10% FBS, 2 nmol/L dihydrotestosterone (DHT), VCaP in DMEM, 20% FBS, 2 nmol/L DHT, and WPMY-1 in DMEM, 10% FBS. LNCaP-NE and LNCaP AI cells were obtained from LNCaP cells cultivated in red-phenol-free RPMI and 10% cs-FBS (charcoal/dextran-treated serum) for 2 weeks to 1 month (NE), or more than 6 months (AI).

### 4.6. Western Blot

Cells (5 × 105) were seeded in 6-well plates and incubated 24 h for adhesion. Cells were lysed in RIPA buffer and proteins were processed for Western blotting, as previously described [37]. Primary antibodies used were: anti-nucleolin (sc-8031, Santa-Cruz Biotechnology, Heidelberg, Germany, 1:500) and β-tubulin (ab6046, Abcam, 1:1000). Immune complexes were detected by chemiluminescence detection with Pierce ECL Western blotting substrate (ThermoFischer Scientific, Courtaboeuf, France) using GBox systems (Syngene, Cambridge, UK).

### 4.7. Statistical Analysis

Statistical analyses were performed by ANOVA unpaired t test using the GraphPad Prism 4.0 software (San Diego, CA, USA). Values of *p* < 0.05 were considered significant. Results were expressed as mean ± SD of at least three determinations for each test from three independent experiments.

## Figures and Tables

**Figure 1 ijms-23-04491-f001:**
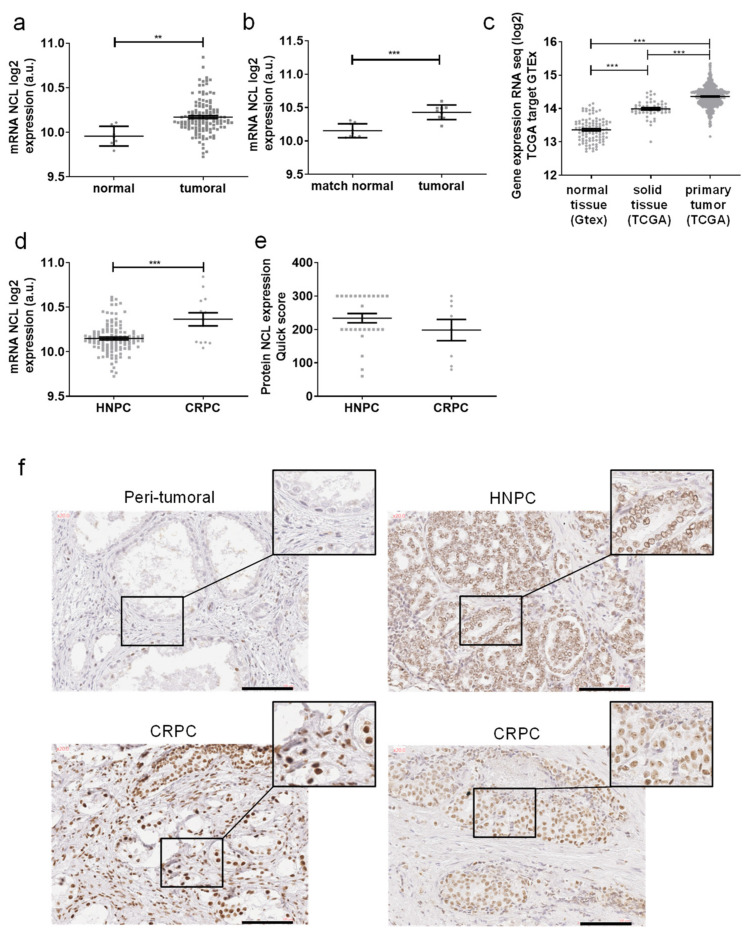
*Nucleolin* (*NCL*) expression in PCa tissues. Gene expression of *NCL* in prostate tissues as measured (**a**) in 9 normal tissues and 130 PCa by HTA2.0 array, (**b**) in 9 paired normal vs. tumoral tissues by HTA2.0 array, (**c**) in GTEx and PRAD TCGA data, (**d**) in 13 CRPC vs. 117 HNPC by HTA2.0 array. (**e**,**f**) NCL protein expression was evaluated by immunochemistry on formalin-fixed paraffin-embedded prostate tissues (27 HNPC and 10 CRPC). (**e**) Staining quantifications of NCL expression using quick scores defined by the intensity and the percentage of stained cells. Scores were analyzed by a pathologist. (**f**) Representative staining of NCL on peritumoral, hormono-naïve prostate cancer (HNPC) and castrate-resistance prostate cancer (CRPC) tissues. Scale bars, 100 µm. Statistical analysis: ** *p* < 0.01, *** *p* < 0.001.

**Figure 2 ijms-23-04491-f002:**
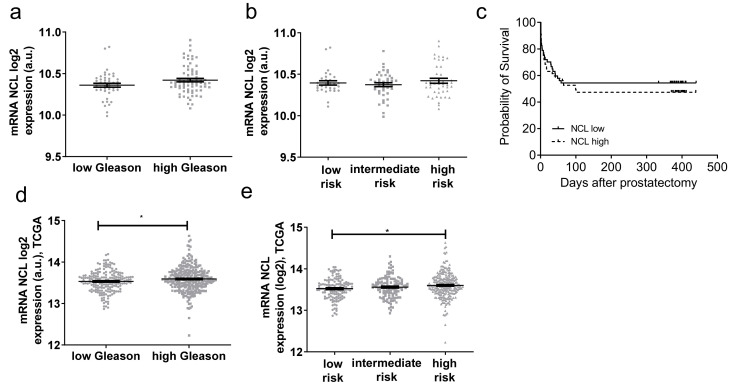
Prognosis value of mRNA *NCL* expression in PCa. Gene expression of *NCL* in prostate tissues measured by HTA2.0 array in PP (**a**–**c**) and TCGA cohorts (**d**,**e**). (**a**,**d**) in low (6 and 3 + 4) versus high (4 + 3, 8, 9) Gleason scores (**b**,**e**) in risk groups and (**c**) progression-free survival. Statistical analysis: * *p* < 0.05.

**Figure 3 ijms-23-04491-f003:**
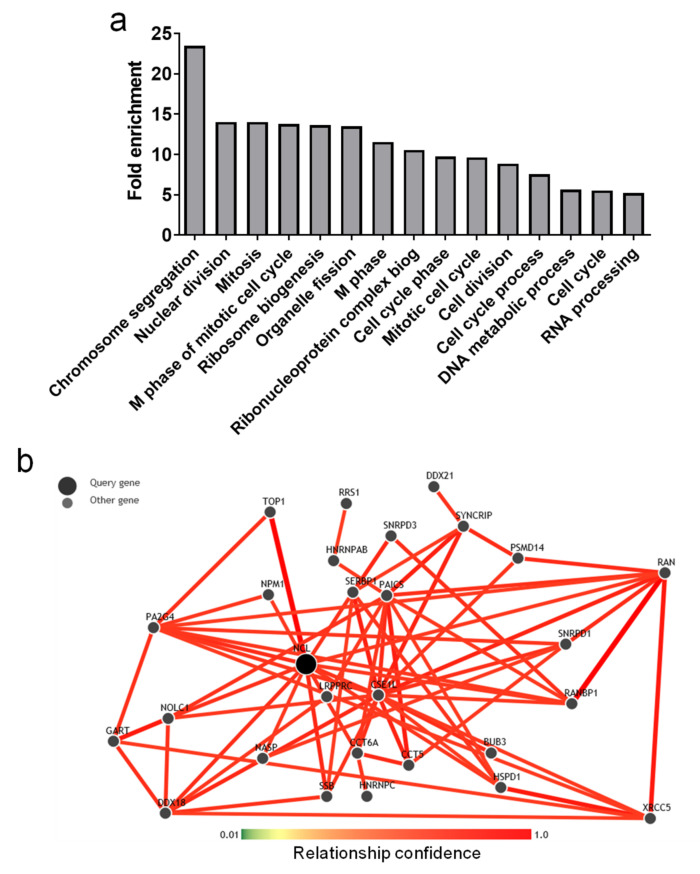
Profile of genes displaying a positive correlation with nucleolin expression. (**a**) Main enriched biological themes (GO terms) defined by DAVID analysis, (**b**) GIANT analysis of correlated gene expression with NCL expression.

**Figure 4 ijms-23-04491-f004:**
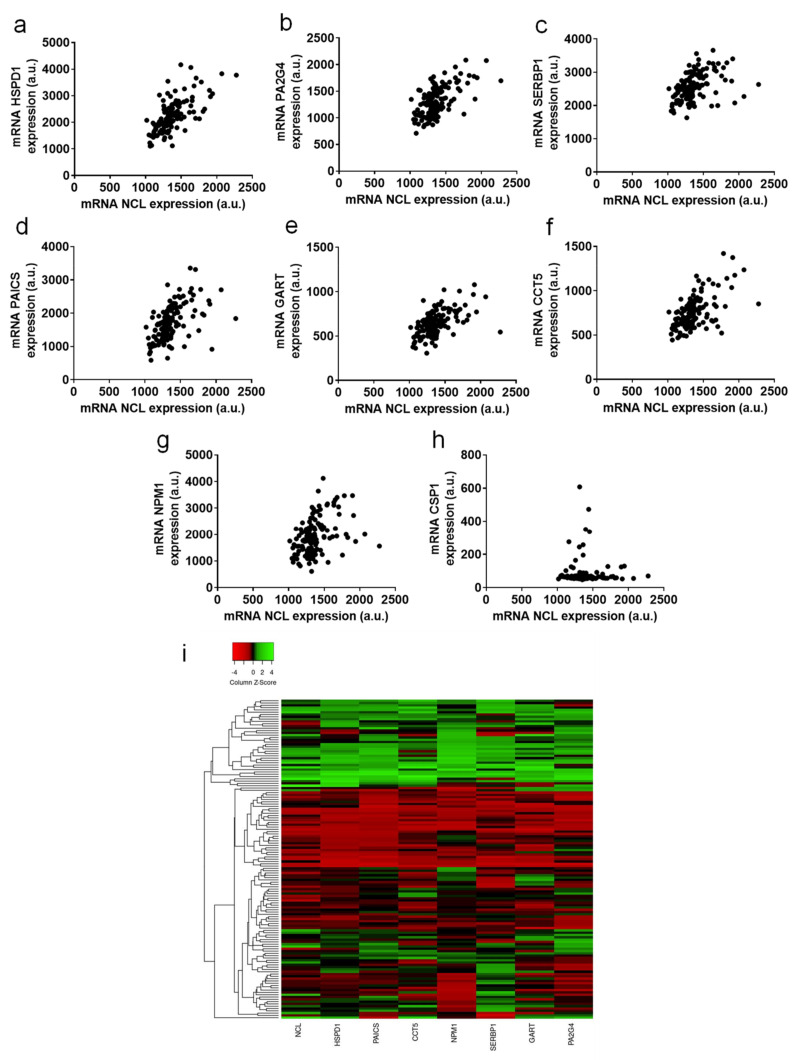
Genes displaying correlation with *nucleolin* expression in PCa. Correlation data between *NCL* expression measured by HTA2.0 array and (**a**) *HSPD1*, (**b**) *PA2G4*, (**c**) *SERBP1*, (**d**) *PAICS*, (**e**) *GART*, (**f**) *CCT5*, (**g**) *NPM1*, and (**h**) *CSP1*. Hierarchical cluster analysis and a heatmap were generated using NCL expression, as assessed by mRNA expression. In the heatmap, each column represents a gene, and each row represents a patient (**i**).

**Figure 5 ijms-23-04491-f005:**
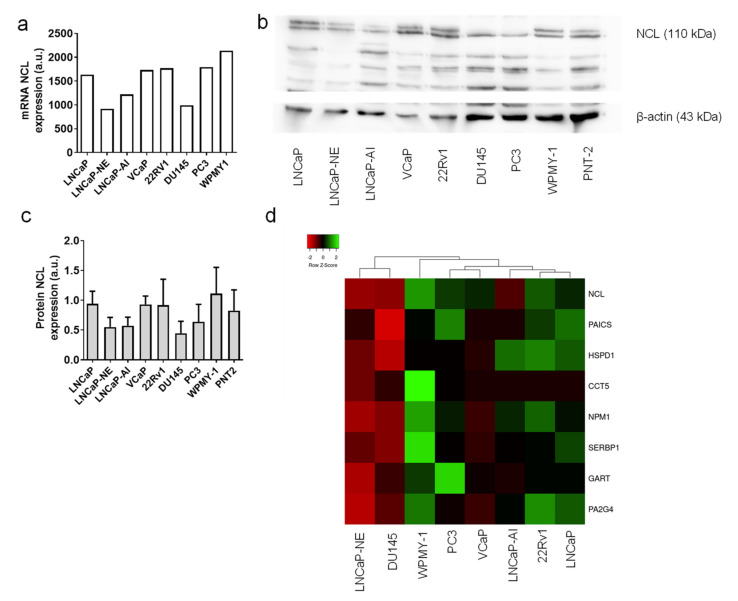
Nucleolin expression in PCa Cell lines. (**a**) NCL mRNA expression in 8 PCa and normal prostate cell lines measured by HTA2.0 array. Expressions are indicated in arbitrary units (a.u.). (**b**,**c**) NCL protein expression in 9 PCa and normal prostate cells lines analyzed by Western blot and quantified by ImageJ software with β-tubulin normalization. Mean ± SD. (**d**) hierarchical cluster analysis and a heatmap were generated using NCL expression, as assessed by mRNA expression in 7 PCa cells lines and an immortalized prostate myofibroblast. In the heatmap, each column represents a different cell line, and each row represents a gene.

**Table 1 ijms-23-04491-t001:** Clinical, biochemical, and pathologic characteristics of the PP cohort. n = 116.

Characteristic	Mean/Median (Range)	No. of Patients (%)
Age, y	62.87/63 (44–76)	
PSA, ng/mL	11.16/7 (1.7–76)	
Follow-up n = 113, mo	99.3/105.7 (1.4–173.9)	
Pathological tumor classification		
pT2a/b		9 (7.7%)
pT2c		28 (24.1%)
pT3a		54 (46.5%)
pT3b		19 (16.3%)
pT4		6 (5.1%)
Prostatectomy gleason score		
6		20 (17.2%)
7 (3 + 4)		25 (21.5%)
7 (4 + 3)		40 (34.5%)
8		26 (22.4%)
9		5 (4.3%)
Relapse n = 113		
yes		56
no		57

**Table 2 ijms-23-04491-t002:** Gene expression in relation with nucleolin expression in PCa. Correlation between *HSPD1*, *PAICS*, *CCT5*, *NPM1*, *SERBP1*, *GART*, *PA2G4*, and *NCL* expressions were evaluated with TCGA data or HTA2.0 array data. Spearman’s correlation, *p*-value, and GIANT score are represented.

		TCGA		PAIR Prostate		GIANT
Correlated Gene	Cytoband	Spearman’s Correlation	p-Value	Spearman’s Correlation	p-Value	Score
*HSPD1*	2q33.1	0.6890	5.62 x10^-70^	0.6844	<0.0001	0.668
*PAICS*	4q12	0.6780	5.62 x10^-67^	0.5860	<0.0001	0.595
*CCT5*	5p15.2	0.6327	5.88 x10^-56^	0.5402	<0.0001	0.558
*NPM1*	5q35.1	0.6087	8.06 x10^-51^	0.4385	<0.0001	0.657
*SERBP1*	1p31.3	0.5227	1.40 x10^-35^	0.5038	<0.0001	0.691
*GART*	21q22.11	0.5145	2.45 x10^--34^	0.5501	<0.0001	0.534
*PA2G4*	12q13.2	0.5054	5.20 x10^--33^	0.5700	<0.0001	0.708
*PAICS*	4q12	0.6780	5.62 x10^--67^	0.5860	<0.0001	0.595
*CCT5*	5p15.2	0.6327	5.88 x10^--56^	0.5402	<0.0001	0.558

## Data Availability

HTA2.0 data have been deposited to the NCBI Gene Expression Omnibus (GSE200879).

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
