# Peer review of "Overexpression of Nucleolin and Associated Genes in Prostate Cancer"

_ijms, 2022, doi:10.3390/ijms23094491_

Round 1
Reviewer 1 Report
The authors did a great job presenting the novelty and significance of this research and presented the results clearly and concisely.
I recommend that this manuscript be accepted after minor revisions and once the following points are addressed:
- Fig 1 e caption should be more descriptive and clear to the readers.
- The immunohistochemistry protocols near to be clearly outlined in section 4.4 and backed up with literature references. Any specific optimizations made must be clearly indicated.
- Figure 4 for the heat map the caption states that "
n the heat map, each column represents a different patient, and each row represents a gene." this should be included in the heat ap graphic.The heat map comparisons are not legible in their current form, it is advised that the orientation of Figure 4 be reconsidered.
Author Response
We would like to thank the reviewer for its comments and suggestions. As required, we have enclosed a detailed list for the changes, point by point. All changes are highlighted in yellow in the main text for better reading.
- Fig 1 e caption should be more descriptive and clear to the readers.
The Fig 1 caption has been changed as recommended to be clearer:
“(e and f) NCL protein expression was evaluated by immunochemistry on formalin-fixed paraffin-embedded prostate tissues (27 HNPC and 10 CRPC). (e) Staining quantifications of NCL expression using quick scores defined by the intensity and the percentage of stained cells. Scores were analyzed by a pathologist. (f) Representative staining of NCL on peri-tumoral, hormono-naïve prostate cancer (HNPC) and castrate-resistance prostate cancer (CRPC) tissues. Scale bars, 100 µm.»
- The immunohistochemistry protocols near to be clearly outlined in section 4.4 and backed up with literature references. Any specific optimizations made must be clearly indicated.
The 4.4 section has been improved as recommended:
“Immunohistochemistry analysis was performed as previously described with minor modifications (Destouches et al., 2016). Formalin-fixed paraffin-embedded (FFPE) tissues were sectioned at 5 µm thickness, deparaffinized and rehydrated. Antigen were unmasked by heat retrieval with pH 9 EDTA buffer for 15 min and endogenous peroxidase activity was inactivated with a 3% hydrogen peroxide solution for 10 minutes. Tissues were then immuno-stained overnight at 4°C with anti-nucleolin antibody (sc-8031, Santa-Cruz, 1:50) diluted in antibody diluent (Zytomed). Immuno-complexes were revealed using the anti-mouse Polink HRP mouse kit and the DAB substrate (Diagomics) with an incubation of 30 minutes at room temperature. Tissues were then stained with hematoxylin and dehydrated. Slides were mounted using Eukitt medium. Protein expression was scored as null (0), weak (1), moderate (2) and strong (3) and the percentage of tumor cells stained was noted. The multiplication of the score and the percentage of stained tumor cells gave the quick score (between 0 and 300). Analysis was performed separately by one genitourinary pathologist (MN) and by two scientific investigators (DD and VF).”
- Figure 4 for the heat map the caption states that "
n the heat map, each column represents a different patient, and each row represents a gene." this should be included in the heat ap graphic.The heat map comparisons are not legible in their current form, it is advised that the orientation of Figure 4 be reconsidered.
The orientation of the Figure 4 has been modified as suggested and legends were also modified: “ Hierarchical cluster analysis and a heat map were generated using NCL expression, as assessed by mRNA expression. In the heat map, each column represents a gene, and each row represents a patient (i)”
Reviewer 2 Report
In the study, the authors examined data on NCL expression in Prostate cancer (PCa) patients and found that nucleolin (NCL) was overexpressed in PCa tissue compared to normal tissue, but did not show prognostic values. Nine genes were highly coexpressed with NCL in patient tissues and tumor prostate cell lines. The authors claim that NCL is an interesting diagnostic biomarker and propose a signa- 26 ture of genes co-expressed with NCL. I think that this paper is well written except for a few minor points of experimental data by western blotting. Specific comments are as follows.
Major points:
- Related to Figure 1 and 2: If possible, it would be better to give a few examples of the expression of NCL in the western blotting in PCa cancerous and surrounding non-cancerous areas.
- Figure 5b: Many nonspecific bands were observed, making me wonder if the antibodies are working properly. Researchers often have trouble with non-specific bands for Santa Cruz antibodies, so I recommend trying another antibody. I think the results of this experiment are very important for this paper.
- English should be carefully revised by a native English speaker or a professional English editing service.
Author Response
We would like to thank the reviewer for its comments and suggestions. As required, we have enclosed a detailed list for the changes, point by point. All changes are highlighted in yellow in the main text for better reading.
Major points:
- Related to Figure 1 and 2: If possible, it would be better to give a few examples of the expression of NCL in the western blotting in PCa cancerous and surrounding non-cancerous areas.
Thank for this very interesting suggestion that would have improved the manuscript. Unfortunately, all the tissues were used for RNA extraction. So, it wasn’t possible to run Western blot analysis as requested.
- Figure 5b: Many nonspecific bands were observed, making me wonder if the antibodies are working properly. Researchers often have trouble with non-specific bands for Santa Cruz antibodies, so I recommend trying another antibody. I think the results of this experiment are very important for this paper.
Indeed, some Santa Cruz antibodies are known to exhibit unspecific bands. Here, the observed NCL bands were described as cleaved forms of NCL protein. These forms are described in different datasheets for NCL antibodies (https://www.thermofisher.com/antibody/product/Nucleolin-Antibody-Polyclonal/PA5-19508, https://www.abcam.com/nucleolin-antibody-ab22758.html#lb, https://www.scbt.com/fr/p/c23-antibody-ms-3) or in some articles (Fang et al., 1993; Kito et al., 2003; Soundararajan et al., 2009), even if the majority of published data shows only the 100-110 kDa bands.
Fang SH, Yeh NH. The self-cleaving activity of nucleolin determines its molecular dynamics in relation to cell proliferation. Exp Cell Res. 1993 Sep;208(1):48-53. doi: 10.1006/excr.1993.1221. PMID: 7689479.
Kito S, Shimizu K, Okamura H, Yoshida K, Morimoto H, Fujita M, Morimoto Y, Ohba T, Haneji T. Cleavage of nucleolin and argyrophilic nucleolar organizer region associated proteins in apoptosis-induced cells. Biochem Biophys Res Commun. 2003 Jan 24;300(4):950-6. doi: 10.1016/s0006-291x(02)02942-x. PMID: 12559966.
Soundararajan S, Wang L, Sridharan V, Chen W, Courtenay-Luck N, Jones D, Spicer EK, Fernandes DJ. Plasma membrane nucleolin is a receptor for the anticancer aptamer AS1411 in MV4-11 leukemia cells. Mol Pharmacol. 2009 Nov;76(5):984-91. doi: 10.1124/mol.109.055947. Epub 2009 Aug 5. PMID: 19657047; PMCID: PMC2774992.
To confirm that these bands aren’t due to the used antibody, we performed a new Western blot comparing the Santa Cruz antibody with the Abcam ab22758 antibody. As it can be observed in the following figure (see word file), the two antibodies give similar results.
Figure: NCL expression in prostate cell lines by Western blot. A: with anti-nucleolin MS3 from Santa Cruz as primary antibody. B: with anti-nucleolin ab22758 from Abcam as primary antibody.
Since this new Western blot using the Santa Cruz antibody exhibits few backgrounds or unspecific staining, we use it in the revised version of the manuscript.
- English should be carefully revised by a native English speaker or a professional English editing service.
Thank you, the English has been reviewed by a native English speaker.

Round 2
Reviewer 2 Report
The authors have addressed almost all of my concerns and I have no further points.